# Evaluation of the Brewing Characteristics, Digestion Profiles, and Neuroprotective Effects of Two Typical Se-Enriched Green Teas

**DOI:** 10.3390/foods11142159

**Published:** 2022-07-21

**Authors:** Yuanyuan Ye, Jiangling He, Zhijun He, Na Zhang, Xiaoqing Liu, Jiaojiao Zhou, Shuiyuan Cheng, Jie Cai

**Affiliations:** 1National R&D Center for Se-Rich Agricultural Products Processing, Hubei Engineering Research Center for Deep Processing of Green Se-Rich Agricultural Products, School of Modern Industry for Selenium Science and Engineering, Wuhan Polytechnic University, Wuhan 430023, China; yyyuan0127@163.com (Y.Y.); hezj@email.szu.edu.cn (Z.H.); 12617@whpu.edu.cn (N.Z.); 18871027325@163.com (X.L.); jiaojiaozhou@whpu.edu.cn (J.Z.); s_y_cheng@sina.com (S.C.); 2Key Laboratory for Deep Processing of Major Grain and Oil, Ministry of Education, Hubei Key Laboratory for Processing and Transformation of Agricultural Products, Wuhan Polytechnic University, Wuhan 430023, China; 3Shenzhen Key Laboratory of Marine Biotechnology and Ecology, College of Life Sciences and Oceanography, Shenzhen University, Shenzhen 518055, China; 4Hubei Key Laboratory of Nutritional Quality and Safety of Agro Products, Wuhan 430064, China

**Keywords:** Se-enriched tea, chemical component, antioxidant activity, dynamic in vitro digestion, neuroprotective effect

## Abstract

As a functional beverage, selenium (Se)-enriched green tea (Se-GT) has gained increasing popularity for its superior properties in promoting health. In this study, we compared the brewing characteristics, in vitro digestion profiles, and protective effects on neurotoxicity induced through the amyloid-beta (Aβ) peptide of two typical Se-GTs (Enshi Yulu (ESYL) and Ziyang Maojian (ZYMJ), representing the typical low-Se green tea and high-Se green tea, respectively). ESYL and ZYMJ showed similar chemical component leaching properties with the different brewing methods, and the optimized brewing conditions were 5 min, 90 °C, 50 mL/g, and first brewing. The antioxidant activities of the tea infusions had the strongest positive correlation with the tea polyphenols among all of the leaching substances. The tea infusions of ESYL and ZYMJ showed similar digestive behaviors, and the tea polyphenols in the tea infusions were almost totally degraded or transferred after 150 min of dynamic digestion. Studies conducted in a cell model of Alzheimer’s disease (AD) showed that the extract from the high-Se green tea was more effective for neuroprotection compared with the low-Se green tea. Overall, our results revealed the best brewing conditions and digestion behaviors of Se-GT and the great potential of Se-GT or Se-enriched green extract (Se-GTE) to be used as promising AD-preventive beverages or food ingredients.

## 1. Introduction

Tea, a beverage generally produced by steeping *Camellia sinensis* leaves, is one of the most widely and commonly consumed drinks around the world, in light of its distinctive flavor and functional properties [1,2]. Based on the degree of fermentation and level of antioxidants, green (non-fermented), oolong (semi-fermented), and black (fermented) teas make up the majority of the categorization of tea [3]. Among them, the health-promoting effects of consuming green tea have been investigated the most extensively, including its anti-cancer, anti-inflammatory, anti-diabetes, anti-obesity, and anti-neurodegenerative activities [4]. Moreover, most of these health benefits can be ascribed to the major bioactive compounds of green tea, including polyphenols, caffeine, amino acids, polysaccharides, and saponins [5]. In particular, polyphenols, such as catechins, are recognized as the most important bioactive compounds in various types of teas due to their key role in health-promoting and disease-preventing properties [6,7]. Furthermore, these polyphenols in green tea show a much higher level than those in black or oolong tea, which accounts for their antioxidant capacity [7].

Selenium (Se) is an essential biological trace element for humans and other mammals, and has great relevance to the prevention of cancer and cardiovascular diseases [8]. The recommended dietary allowance for Se intake in adults is currently 55 μg per day, with a recommended tolerable upper intake level of 400 μg per day [9]. An inadequate dietary Se intake is detrimental to health and contributes to many diseases such as cancer, coronary heart disease, and Keshan disease [10], affecting 0.5–1 billion people worldwide [11]. However, excessive Se intake can induce toxicity in the human body and lead to symptoms of alopecia, hair and nail abnormalities, skin injury, neurological disorders, and even paralysis and death [12]. Thus, moderate Se supplementation is of great necessity for human health.

The *Camellia sinensis* plant takes up Se directly from the soil and accumulates it in its leaves [13]. Based on the agricultural industry standard of the People’s Republic of China (NY/T 600-2002), the Se content of Se-enriched tea should be 0.25–4 mg/kg. Se-enriched tea is considered as a healthy beverage, in which the combination of Se, the tea polyphenols, and other bioactive compounds has been found to show a potential synergistic effect by enhancing the health-promoting effects [14]. Prior research has suggested that Se-enriched green tea (Se-GT) has higher antioxidant, prebiotic, and anticarcinogenic activities than common tea [15,16]. The pronounced nutritional properties have resulted in an ever-increasing demand for Se-GT in recent years.

It is noteworthy that the health benefits largely depend on the leaching substances from the tea under different brewing conditions [17]. Studies have demonstrated that different antioxidant properties are associated with the difference in the phytochemicals released in tea infusions caused by different brewing conditions, such as brewing time and temperature [18]. Additionally, the quality of the tea infusion is dependent on a variety of factors, including tea type, brewing duration, water temperature, water-to-tea ratio, and brewing time [19]. There are many reports about the impact of brewing methods on the leaching of phytochemicals and antioxidant activities in tea infusions [20,21,22]; however, most studies were performed using common tea instead of Se-enriched tea. Therefore, it is of great significance to investigate the brewing characteristics of Se-GT.

Recently, an in vitro digestion technique has been increasingly employed to assess the bioavailability of individual antioxidants from a variety of foods and dietary supplements, and it is recognized as a true reflection of the potential health effects [23]. Previous digestion studies mainly focused on static in vitro digestion [24,25], whereas only a few studies were based on a dynamic in vitro digestion system. As the dynamic digestion model can mimic the physico-chemical and physiological processes of the in vivo environment, a better digestion performance can be achieved than that of the traditional static in vitro digestion model [26,27]. Furthermore, the amount and activity of the antioxidants present during the digestion of Se-enriched green tea infusion have been scarcely explored. Thus, investigating how the digestion process affects the bioactivity of the Se-GT infusion is of critical importance.

Compared with the relatively large number of studies on the anticarcinogenic potential of Se-enriched tea [14,16], there are currently very few studies on Se-enriched tea regarding the neuroprotective effects on neuronal cells. Alzheimer’s disease (AD) is a progressive and irreversible neurodegenerative disease linked to cognitive disorder [28], and represents one of the greatest health-care challenges of the 21st century, affecting an increasing number of older people worldwide [29]. The accumulation of the amyloid-beta (Aβ) peptide is the primary contributor to AD pathogenesis, and inhibition of the neurotoxicity induced by Aβ may thus serve as a promising therapeutic strategy for the remission of AD onset and progression [30]. Among the tea components, catechins (especially epigallocatechin gallate (EGCG)), L-theanine, and caffeine have been found to effectively reduce Aβ-induced neurotoxicity through various potential mechanisms [31,32,33,34]. However, whether the Se-enriched green tea extract (Se-GTE) could exert neuroprotection against Aβ-induced toxicity has not been investigated to date.

The overall goal of this study was to compare the chemical component (including tea polyphenols, caffeine, free amino acids, soluble sugar, water extracts, and Se) leaching and antioxidant activities of tea infusions with different brewing conditions (including duration, temperature, water-to-tea ratio, and brewing time), in vitro digestion profiles of tea infusions steeped in the optimized condition, and the potential protective effects of tea extracts against Aβ_1–42_-induced neurotoxicity in HT22 cells for two typical Se-GTs. Our results can provide a basis for the scientific brewing of Se-GT and new insight for the better utilization of Se-GT as beverages and functional foods in order to prevent and alleviate chronic neurodegenerative disease.

## 2. Materials and Methods

### 2.1. Materials and Characterizations

The Enshi Yulu (ESYL) green tea was obtained from Enshi Lanbei Tea Co., Ltd. (Enshi, China), and the Ziyang Maojian (ZYMJ) green tea was purchased from Ziyang Hongwei selenium-enriched Agricultural Technology Co., Ltd. (Ankang, China), both with a date of manufacture in 2020. Sinopharm Chemical Reagent Co., Ltd. (Beijing, China) provided the sodium carbonate (Na_2_CO_3_), Folin−Ciocalteu reagent, basic lead acetate, sulphuric acid (H_2_SO_4_), disodium hydrogen phosphate (Na_2_HPO_4_), potassium dihydrogen phosphate (KH_2_PO_4_), ninhydrin, potassium persulfate (K_2_S_2_O_8_), nitric acid (HNO_3_, GR), hydrochloric acid (HCl, GR), sodium hydroxide (NaOH, GR), absolute ethanol, potassium chloride (KCl), potassium dihydrogen phosphate (KH_2_PO_4_), sodium bicarbonate (NaHCO_3_), sodium chloride (NaCl), magnesium chloride hexahydrate (MgCl_2_(H_2_O)_6_), ammonium carbonate ((NH_4_)_2_CO_3_), and calcium chloride (CaCl_2_). Gallic acid (GA) was purchased from Yien Chemical Technology Co., Ltd. (Shanghai, China). Caffeine was obtained from Tanmo Quality Inspection Reference Material Center (Beijing, China). The plant soluble sugar content detection kit was purchased from Solarbio Science and Technology Co., Ltd. (Beijing, China). The selenium (Se) standard solution was purchased from the National Nonferrous Metals and Electronic Materials Analysis and Testing Center (Beijing, China). The potassium borohydride (KBH_4_) was purchased from Kemiou Chemical Reagent Co., Ltd. (Tianjin, China). 2,2-Diphenyl-1-picrylhydrazyl (DPPH), 2,2′-azinobis(3-ethylbenzothiazoline-6-sulfonic acid ammonium salt) (ABTS), stannous chloride (SnCl_2_ 2H_2_O), pepsin (from gastric porcine mucosa), and pancreatin of the porcine pancreas were purchased from Aladdin Biochemical Technology Co., Ltd. (Shanghai, China). L-Theanine and bile salt from pigs were brought from Macklin Biochemical Co., Ltd. (Shanghai, China). The HT22 cells were obtained from the laboratory of life sciences and oceanography at Shenzhen University (Shenzhen, China). Aβ_1–42_ and dimethyl sulfoxide (DMSO) were purchased from Sigma-Aldrich (St. Louis, MO, USA). Dulbecco’s Modified Eagle Medium (DMEM), fetal bovine serum (FBS), penicillin/streptomycin solution, and phosphate buffered saline (PBS) solution were obtained from Gibco BRL (Grand Island, NY, USA). Cell Counting Kit-8 (CCK-8) was brought from Beyotime Biotechnology (Jiangsu, China). All of the chemicals used were analytical grade, unless otherwise stated. Experimental water was purified on a Milli-Q system (Milllipore, Molsheim, France).

Absorbance was recorded using an EnSpire Multimode Reader (Perkin Elmer, Waltham, MA, USA). Microwave digestion was performed on a Cash Cow 4010 microwave digestion apparatus (Preekem Technology Development Co., Ltd., Shanghai, China). The acid evaporation process was taken on a G-400 graphite digester (Preekem Instrument Technology Development Co., Ltd., Shanghai, China). The Se content was measured on an AFS-8530 system (Haiguang Instrument Co., Ltd., Beijing, China). Dynamic in vitro digestion was conducted in a dynamic human stomach-intestine IV (DHSI-IV) model (Xiao Dong Pro-health Instrumentation Co., Ltd., Suzhou, China).

### 2.2. Preparation of Tea Infusions

After being weighed based on the water-to-tea ratios (30, 40, 50, 60, 70, and 80 mL/g), the tea leaves were brewed with a certain volume of deionized water at the studied temperatures (70, 75, 80, 85, 90, and 95 °C) for 1, 2, 3, 5, 7, and 9 min to obtain the corresponding infusion. Upon completion, the mixture was immediately filtered. The tea residue was collected. Frozen and lyophilized samples were used for the analysis of the Se content, while the cooled infusions were then used to analyze their chemical composition and antioxidant activity. All of the experiments were repeated at least three times in triplicate.

### 2.3. Determination of the Content of Tea Polyphenols, Caffeine, Free Amino Acids, Soluble Sugar, and Water Extracts

Detailed methods are presented in the Appendix A.

### 2.4. Determination of Total Se

The dried samples were powdered and filtered through the 100-mesh sieve. First, 0.15 g of tea powder and 7 mL of nitric acid (HNO_3_) were digested in the microwave digestion apparatus under the recommended digestion parameters. The digestion program was set as follows: the sample was heated to 80 °C and maintained for 3 min, heated to 120 °C and maintained for 3 min, heated to 150 °C and maintained for 3 min, heated to 180 °C and maintained for 3 min, heated to 200 °C and maintained for 20 min, and finally cooled to 55 °C. By using a graphite digester, the mixtures were heated to 200 °C to volatilize the excess nitric acid, and then added with 5 mL of hydrochloric acid (HCl; 50% *v*/*v*) and evaporated to a volume of 1 mL. The resulting solution was transferred to a centrifugal tube, adjusted with HCl (10% *v*/*v*) to 10 mL, and then analyzed using an AFS-8530 system.

### 2.5. Dynamic In Vitro Digestion

#### 2.5.1. Preparation of Simulated Digestion Solutions

The simulated digestion solution was mainly composed of the corresponding electrolyte stock solution, enzymes, CaCl_2_, and water, and the preparation method described by M. Minekus [35] was used, with some minor modifications. The electrolyte stock solution for simulated gastric fluid (SGF) contained KCl (6.9 mM), KH_2_PO_4_ (0.9 mM), NaHCO_3_ (25 mM), NaCl (47.2 mM), MgCl_2_(H_2_O)_6_ (0.1 mM), and (NH_4_)_2_CO_3_ (0.5 mM). The electrolyte stock solution for the simulated intestinal fluid (SIF) contained KCl (6.8 mM), KH_2_PO_4_ (0.8 mM), NaHCO_3_ (85 mM), NaCl (38.4 mM), and MgCl_2_(H_2_O)_6_ (0.33 mM). For the preparation of SGF (take 100 mL as an example), 50 μL of 0.3 M CaCl_2_ and 19.45 mL of water were mixed with 80 mL of SGF electrolyte stock solution, followed by the addition of 3 M HCl to adjust the pH to 1.6, and then pepsin was added to reach 800 U/mL in the final digestion mixture. For the preparation of SIF (taking 100 mL as an example), 200 μL of CaCl_2_ and 19.45 mL of water were mixed with 80 mL of SIF electrolyte stock solution, followed by the addition of 3 M HCl to reduce the pH to 7.0, and then pancreatin and bile salt were added to achieve 200 U/mL and 0.26 g/L in the final digestion mixture.

#### 2.5.2. In Vitro Digestion Using the DHSI-IV Model

In vitro gastroduodenal digestion was conducted in a DHSI-IV model. The digestion process was controlled automatically by the computer, and the system was maintained at 37 °C. Feeding 250 mL of tea infusion samples brewed under optimized conditions (5 min, 90 °C, 1:50) into the esophagus model immediately started the gastroduodenal digestion. The total digestion time was 3 h. The secretion rates of SGF were 30, 1.2, 1.6, 2, 2.5, 2.1, 1.6, and 1.5 mL/min in the time periods of 0–1, 1–10, 10–20, 20–30, 30–40, 40–50, 50–60, and 60–90 min, respectively. The flow rate of SIF was maintained at 1.3 mL/min during the 180 min digestion. Digesta (2 mL) were sampled after 15, 30, 45, 60, 75, and 90 min of gastric digestion and after 15, 30, 45, 60, 75, 90, 105, 120, 135, 150, 165, and 180 min of duodenal digestion. After adding equal volumes of absolute ethanol to inactivate the enzyme and mixing, samples were rapidly frozen in centrifuge tubes at −20 °C until the tea polyphenol content and antioxidant capacity were measured. Three digestion replicates were performed.

### 2.6. Determination of Free Radical Scavenging Activity

The antioxidant capacity of the tea infusions brewed under different conditions and digested samples collected at different time points was determined using 2,2-diphenyl-1-picrylhydrazyl (DPPH) and 2,2′-azinobis(3-ethylbenzthiazoline-6-sulfonic acid (ABTS) assays, as described previously [36,37]. Detailed methods are reported in the Appendix A.

### 2.7. Se-GTE Preparation

Se-GT was extracted (tea/water, *w*/*v* = 1:50) using deionized water at 90 °C, and the extraction time was extended to 45 min for improving the final yield. The filtrates were lyophilized to obtain Se-GTEs, including ESTE (tea extract of ESYL) and ZYTE (tea extract of ZYMJ), followed by storage at −20 °C for further use.

### 2.8. Cell Assay

#### 2.8.1. Preparation of Aβ_1–42_ Oligomer

The Aβ_1–42_ peptide was dissolved initially in DMSO as a stock solution at 1 mM and then stored at −20 °C. For the preparation of the Aβ_1–42_ oligomer, working solutions with a series of concentrations of 5, 10, 20, and 40 μM were prepared by dilution of the stock solution with basic medium and were incubated at 4 °C for 12 h before use.

#### 2.8.2. Cell Culture and Treatment

The HT22 cells were cultured in DEME containing 10% FBS and 1% penicillin/streptomycin at 37 °C in a humid atmosphere with 5% (*v*/*v*) CO_2_. The HT22 cells were plated in 96-well plates for 24 h and exposed to various concentrations of Aβ_1–42_ (5–40 μM) or Se-GTEs (5–100 μg/mL) for another 24 h, after which the cell viability was determined using the CCK-8 assay.

#### 2.8.3. Cell Viability via CCK-8 Assay

The viability of HT22 cells was determined via using the CCK-8 assay. Briefly, HT22 cells were seeded in 96-well plates at a density of 5 × 10^3^ cells/well and were cultured in the complete medium for 24 h. Then, HT22 cells were pretreated with different concentrations of green tea extracts (5, 10, and 20 μg/mL) for 2 h, followed by incubation with Aβ_1–42_ (20 μM) for another 22 h. Finally, the HT22 cells were treated with CCK-8 solutions and were further incubated at 37 °C for 2 h. The absorbance at 450 nm was recorded using a microplate reader. All of the assays were performed in triplicate.

### 2.9. Statistical Analysis

All of the results are reported as mean ± standard deviation (SD) for at least three independent experiments performed in triplicate. GraphPad Prism 9 (GraphPad Software Inc., San Diego, CA, USA) was used for the statistical analyses in the present study. Student’s *t*-test (*p* < 0.05) was used to compare the differences between groups [38].

## 3. Results and Discussion

### 3.1. Chemical Properties of Se-GT

The appearance of the two green tea samples is shown in Appendix A. To confirm the purchased tea was Se-enriched tea, the contents of Se and the main components in ESYL and ZYMJ were determined, as displayed in Table 1. It is worth noting that ZYMJ had a 7.5-fold higher Se concentration (4.33 ± 0.18 μg/g) than that (0.58 ± 0.01 μg/g) of ESYL. The higher contents of tea polyphenols, caffeine, and water extracts were found in ZYMJ, and the higher contents of free amino acids and soluble sugar were found in ESYL.

### 3.2. Effect of Brewing Conditions on the Phytochemical Compositions of Se-GT

Among all of the leaching substances in tea infusions, the dissolution rate, from high to low, was tea polyphenols, free amino acids, caffeine, and soluble sugar, respectively (Figure 1). It can be also seen that ESYL had a lower leaching rate of tea polyphenols and a higher leaching rate of free amino acids, caffeine, and soluble sugar, compared with ZYMJ. The trend of dissolution was nearly identical for both of the tea samples. It was found that the longer the brewing time, the larger the dissolution rate for almost all of the detected phytochemicals (Figure 1A,E). Rapid dissolution was observed for tea polyphenols, caffeine, free amino acids, soluble sugar, and water extracts within the first 5 min, and then the release rate slowed down over the subsequent 4 min, finally reaching the maximum concentration at 9 min. The above substances (except for the free amino acids) also showed an initial increase in dissolution followed by a decrease, with a subsequent maximum at a temperature of 90 °C (Figure 1B,F). The higher brewing temperature resulted in increased leaching of the free amino acids in both tea samples, with maximum dissolution rates at 95 °C. The results for the effects of time and temperature were similar to those from previous studies [39,40]. Tea polyphenols, caffeine, and soluble sugar leaching into the tea infusions first increased, then decreased, and finally increased with the increase in the water-to-tea ratio (Figure 1C,G). The free amino acids and water extract dissolution level, on the contrary, exhibited the opposite trend, reaching maximums at 30 mL/g and 70 mL/g, respectively. It is also worth noting that all inflection points were observed at a 50 mL/g scale. The leaching of all of the detected phytochemicals during the three times of brewing in ESYL and ZYMJ was determined and is shown in Figure 1D,H. The concentrations of the main chemicals in the green tea infusions dramatically decreased with the increased brewing times. Furthermore, the release of the compounds investigated in the third brewed tea infusions was only about 1/3–1/4 of that in the first brewed tea infusions. This could be attributed to the fact that ESYL and ZYMJ are green teas with a high raw material tenderness and a fast-leaching rate of chemical components, and are thus not suitable for multiple brewing [19].

The longer brewing time at a higher temperature had a negative impact on the sensory quality of the tea infusion [19]. Water with an extremely high temperature during the tea brewing process might yield massive amounts of bitter and astringent compounds such as catechins to be incorporated into green tea, which could increase the bitterness and astringency of green tea [41]. Meanwhile, brewing green tea for a long time also results in green tea with a much more bitter and more astringent mouthfeel [42]. Although the continuous brewing of tea will still have leaching of chemical components, it will lead to a decline in sensory quality to a certain extent [43]. Therefore, our results suggest that a time of 5 min, temperature of 90 °C, water-to-tea ratio of 50 mL/g, and first brewing are the best combinations for the leaching of phytochemicals and for obtaining a better taste of Se-GT.

### 3.3. Effect of Brewing Conditions on the Se Leaching of Tea Infusions

In order to investigate the impact of brewing conditions on the dissolution characteristics of Se in Se-GT, the Se content of the tea leaves before and after brewing was measured and the dissolution rate of Se was calculated. According to the leaching rate of Se shown in Figure 2, it was evident that the tea infusion of ZYMJ had a higher Se content than that of ESYL, which was in line with the Se content in the tea leaves. Moreover, essentially the same leaching trends were observed in both ESYL and ZYMJ with different brewing conditions. The leaching rate of Se increased with the brewing time and brewing temperature (Figure 2A,B). The dissolution level of Se in ESYL steadily rose as the water-to-tea ratio increased from 30 mL/g to 80 mL/g, and reached a maximum (16.10%) at 80 mL/g (Figure 2C). However, with the gradual increase in the water-to-tea ratio, ZYMJ showed an initial increase in the Se dissolution rate followed by a decrease, and then peaked at 15.63% when the ratio was 80 mL/g. As displayed in Figure 2D, the leaching rate of Se in the tea infusions from the first brewing was greater than that of multiple brewing, which is in line with the previously reported results [44].

### 3.4. Effect of Brewing Conditions on the Antioxidant Activity of Tea Infusions

The health benefits of tea are mostly based on its antioxidant activity [37]. The antioxidant capacities of the Se-GTs infusions prepared with different brewing conditions are summarized in Figure 3, which shows a strong dependence on the brewing conditions. It was observed that the ABTS and DPPH antioxidant scavenging abilities of the ZYMJ infusion were mostly much higher than those of the ESYL infusion. According to the DPPH and ABTS assays, a longer brewing time and higher brewing temperature resulted in an increasing antioxidant capacity, while a higher water-to-tea ratio and longer brewing time resulted in a weaker antioxidant capacity. These results were consistent with some previous findings [45,46]. Gan, P.T. and Ting, A.S.Y [47] reported that the antioxidants activities increased with the infusion time, but decreased with the three brewing cycles. Such changes in antioxidative activity also showed rapid and slow varying phases. This observation was in line with previous reports [48,49]. Furthermore, it was found that the highest antioxidant capacity could be obtained at time of 5 min, temperature of 90 °C, and water-to-tea ratio of 30 mL/g of first-brew infusions of ZYMJ and ESYL.

### 3.5. Correlation Analysis

We analyzed the correlations between the leaching substances and antioxidant activities of the tea infusions, the results of which are shown in Appendix A. Tea polyphenols had the highest positive correlation with the antioxidant activity among all of the assessed chemicals (Appendix A). This result is logical and reasonable as polyphenols are known to be the major antioxidant constituents in tea [50,51]. For caffeine, there were moderate positive correlations with the DPPH (R^2^ = 0.4367) or ABTS (R^2^ = 0.4051) scavenging abilities (Appendix A). These results agree well with those of a previous study, which showed that the antioxidant activity in tea contributed by caffeine is less important than those contributed by the phenolic compounds [37]. Although the leaching rate of the free amino acids in the tea infusion were next to those of the tea polyphenols, it was moderately positively correlated with the DPPH (R^2^ = 0.4350) scavenging abilities and weakly positively correlated with the ABTS (R^2^ = 0.3638) scavenging abilities (Appendix A). It can be considered that the presence of amino acids could serve a promoting role in the antioxidant system [52]. It was also found that soluble sugar and Se in the tea infusion exhibited weak positive associations with the antioxidant activity (Appendix A). Furthermore, Appendix A shows a moderate positive relationship between the water extract dissolution level and the antioxidant capacity. Therefore, these results demonstrated that tea polyphenols were the main contributors to the antioxidant properties of the Se-GT infusion.

### 3.6. Polyphenol Release and Antioxidant Activity during In Vitro Gastroduodenal Digestion

Appendix A shows the structure of the DHSI-IV model. Figure 4 shows the changes in the concentration of the tea polyphenols and the level of antioxidant activities of the digesta during the simulated Se-GT infusion gastroduodenal digestion. The fact that the concentration of tea polyphenols in the digesta decreased over time during the simulated gastric digestion suggested that the polyphenols in the tea infusion were degraded (Figure 4A). Such a reduction was more obvious during the first 15 min, which could be related to the abundant digestive juices secreted in the soft stomach model [53]. After 90 min of gastric digestion, the concentrations of residual tea polyphenols in the stomach were 0.37 mg/mL (ZYMJ) and 0.32 mg/mL (ESYL). Furthermore, the levels of tea polyphenols for ZYMJ during the gastric phase were consistently higher than those for ESYL. One potential cause of this discrepancy might be the levels of polyphenols contained in the tea leaves. For the duodenal digestion, our first detection time point was 15 min post-digestion. Subsequently, the tea polyphenol concentration in the duodenal digesta first rose and reached the maximum concentration after digestion for 45 min, with 0.68 mg/mL (ZYMJ) and 0.56 mg/mL (ESYL). Such an increase in tea polyphenols could be attributed to the greater accumulation of digesta from the stomach in the duodenum. As digestion progressed further, the polyphenol content in the duodenal digesta started to gradually decline within 45–150 min, and eventually reached a relatively stable level of all around 0.03 mg/mL for the digestive samples. These data also suggest that the tea infusions of ZYMJ had a relatively better performance in the tea polyphenol digestion and absorption in the gastroduodenal environment compared with ESYL.

The DPPH and ABTS of the digested mixture had similar changes to the tea polyphenols throughout the gastroduodenal digestion period, although the antioxidant activity value exhibited varying degrees of reduction as the digestion progressed, as shown in Figure 4B,C. In addition, the digesta of the ZYMJ tea infusion had consistently higher antioxidant capacity than that of the ESYL. The DPPH and ABTS scavenging rates showed three main patterns: increasing at the initiation of duodenal digestion (15–45 min), rapidly declining (45–150 min), and remaining relatively stable until the end of digestion. The DPPH scavenging rate for ZYMJ showed a value of 17.00% at the end of digestion, which was much higher than that for ESYL (11.15%), indicating a greater level of residual active components in ZYMJ. The majority of loss in the antioxidant activity of the digested samples reflected by the DPPH values occurred during duodenal digestion, while a greater extent of reduction in ABTS radical scavenging capacity was observed in the gastric digestion phase. Thus, digestion reduced the antioxidant activity in all of the tested tea infusions and a greater free radical scavenging was observed for the digesta of ZYMJ.

It is noteworthy that antioxidative capacity is associated with changes in the polyphenol levels and pH values [24,54]. It was previously shown that the transition from the gastric to the intestinal environment results in slight structural changes to the polyphenols [23]. Dietary polyphenols are highly sensitive to weak alkaline conditions in the intestine, and a large portion of these compounds can be converted into other unknown and/or undetected structural forms with different chemical properties and biological activity after in vitro digestion [55]. Polyphenols are vulnerable to degradation when incubated at an elevated pH (6–8) [56]. These analyses further suggest that our results are in line with the digestion properties of polyphenols.

### 3.7. Effect of Se-GTEs on Viability of HT22 Cells against Aβ_1–42_ Induced Toxicity

We first assessed the toxic effect of Se-GTEs on HT22 cells and found that the cell viability was not significantly decreased by low concentrations of Se-GTEs (5–20 μg/mL), but it was significantly reduced by the introduction of more Se-GTEs (25–100 μg/mL) (Figure 5A,B), indicating that Se-GTEs at a concentration of <20 μg/mL cannot obviously induce cytotoxicity. The significant results shown in Figure 5A,B also revealed that ESTE could induce a relatively higher cytotoxicity than ZYTE. Aβ_1–42_ has a strong neurotoxicity and can lead to neuronal apoptosis [57]. HT22 cells were treated with different doses of Aβ_1–42_ (0, 5, 10, 20, and 40 μM) for 24 h and the cell viability was measured. The result of the CCK-8 assay demonstrated that Aβ_1–42_ led to a concentration-dependent decrease in HT22 cell viability and the cell survival rate was significantly reduced to 43.40% at a concentration of 20 μM (Figure 5C). Thus, 20 μM of Aβ_1–42_ was used for the subsequent experiments. Thereafter, the effects of Se-GTEs on the viability of the HT22 cells against Aβ_1–42_-induced toxicity were investigated. Data from Figure 5D,E show that Se-GTEs pretreatment enhanced the viability of Aβ_1–42_-induced cells in a dose-dependent manner. No significant increase in cell viability was observed until the concentration of pretreated ESTE reached 20 μg/mL, while pre-incubating HT22 cells with ZYTE at concentrations of 10–20 μg/mL significantly attenuated the cytotoxic effects of Aβ_1–42_. It can be deduced that the protective effect of ESTE against Aβ_1–42_-induced cell death was less pronounced than that of ZYTE. The beneficial effects of Se-GTEs were also confirmed by microscopic observations (Appendix A). It was shown that Aβ_1–42_-induced cells exhibited reduced cell numbers and were significantly impaired with visible shrinkage, suspension, and irregular shapes compared with the control cells. Consistent with previous cell viability results, it was observed that the morphology of the cells pretreated with different concentrations of Se-GTEs (10 μg/mL and 20 μg/mL) was markedly improved, and the cell numbers were also correspondingly increased when compared with those of the Aβ_1–42_ group. These results strongly suggest that Se-GTEs could mitigate Aβ_1–42_-induced neurotoxicity at a certain concentration, and ZYTE showed a better protective effect than ESTE.

## 4. Conclusions

In the selected samples, there was a higher content of tea polyphenols and Se in ZYMJ than in ESYL. The chemical compositions (including tea polyphenols, caffeine, free amino acids, soluble sugar, water extracts, and Se) and antioxidant activities in the tea infusions of ESYL and ZYMJ exhibited similar trends as the brewing conditions changed, and the optimized brewing parameters were 5 min, 90 °C, 50 mL/g, and one time of tea brewing. The antioxidant capacity of Se-GT infusions was positively correlated with tea polyphenols. The gastroduodenal digestive behavior of the Se-GT infusions was investigated in the DHSI-IV model, and the dynamics of the antioxidant activities during digestion were found to be consistent with that of the tea polyphenol concentration. All of the Se-GT samples were rapidly digested in the first 90 min, and the levels of tea polyphenols and antioxidant activities became relatively stable after 150 min of digestion. Moreover, it was observed that pretreatment with Se-GTE could effectively mitigate neurotoxicity induced by Aβ_1–42_ in HT22 cells, and the tea extract from green tea with higher tea polyphenols and a higher Se content presented a stronger neuroprotective capacity compared with that from green tea with lower tea polyphenols and a lower Se content. Notably, our comparative evaluation cannot represent the geographical differences as a whole. This may be the first study to directly evaluate the neuroprotection of Se-GTE. In summary, this work provides powerful theoretical grounds for the scientific brewing of Se-GT and some initial evidence for the neuroprotective role of Se-GTE. The novel Se-enriched green tea product may be a potential food ingredient for the development of various functional foods and for the treatment of chronic inflammation-related diseases. Future work should be directed at identifying the specific mechanisms of action of Se-GTE.

## Figures and Tables

**Figure 1 foods-11-02159-f001:**
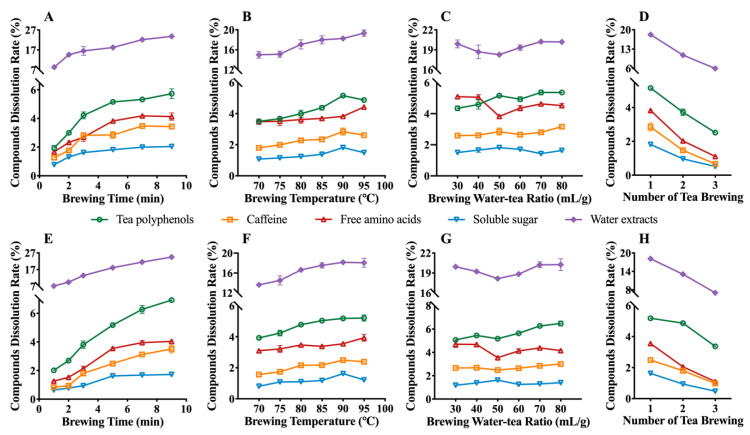
Effects of brewing conditions on the dissolution rate of the phytochemical compositions of tea infusions for ESYL (**A**–**D**) and ZYMJ (**E**–**H**). Data are shown as mean ± SD from the triplicate analysis.

**Figure 2 foods-11-02159-f002:**
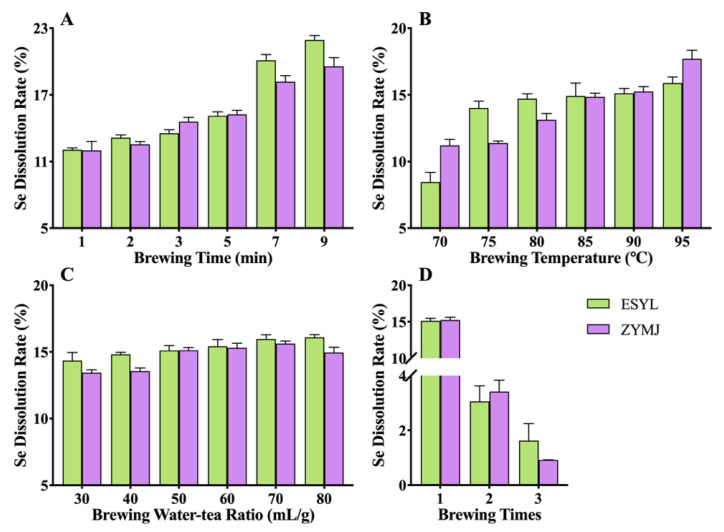
Effects of brewing conditions (brewing time (**A**), brewing temperature (**B**), brewing water-tea ratio (**C**), and brewing times (**D**)) on Se leaching from Se-GT leaves (ESYL and ZYMJ). Data are presented as mean ± SD of the triplicate experiments.

**Figure 3 foods-11-02159-f003:**
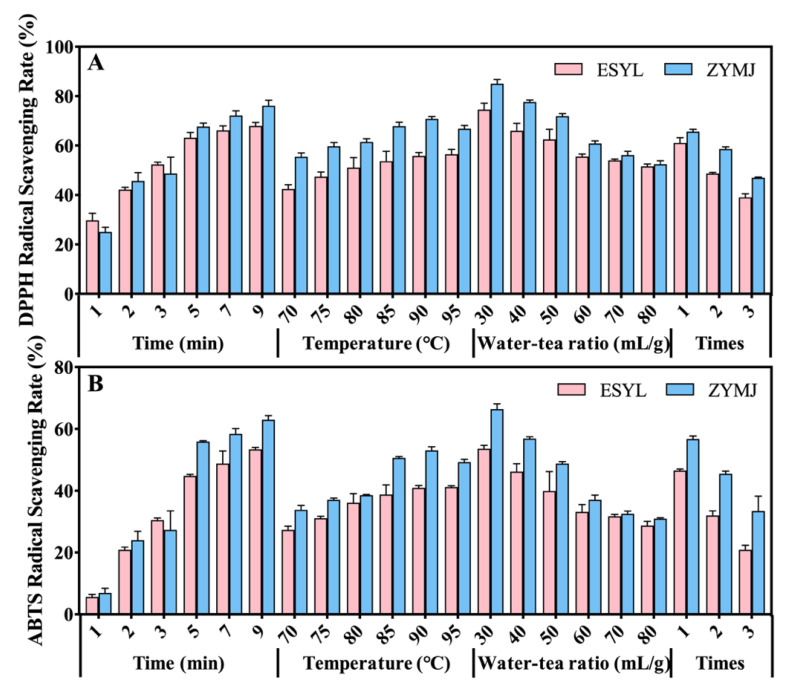
Effects of brewing conditions on the antioxidant activity of tea infusions for ESYL and ZYMJ. The antioxidant activity of green tea infusions determined using DPPH (**A**) and ABTS (**B**) assays. Data are expressed as mean ± SD of the triplicate experiments.

**Figure 4 foods-11-02159-f004:**
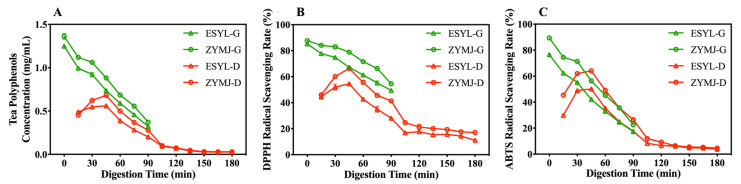
Changes in the tea polyphenol concentration (**A**) and antioxidant activity (DPPH (**B**) and ABTS (**C**)) of the digesta at various stages during the simulated gastroduodenal digestion in vitro. Values are means ± SD from three independent triplicate experiments. G, simulated gastric digestion stage; I, simulated duodenal digestion stage.

**Figure 5 foods-11-02159-f005:**
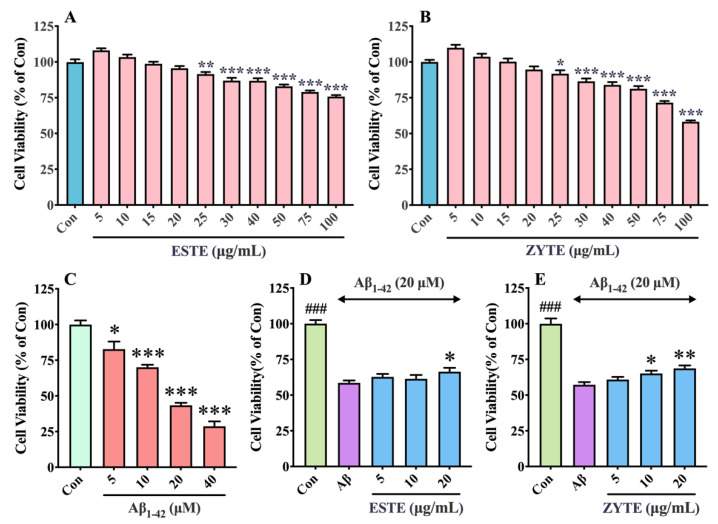
Effects of Se-GTEs on Aβ_1–42_-induced cytotoxicity in HT22 cells. HT22 cells were treated with different concentrations of Se-GTEs (**A**,**B**) or Aβ_1–42_ (**C**) for 24 h. HT22 cells were treated with 5–20 μg/mL of Se-GTEs for 2 h, followed by incubation with 20 μM of Aβ_1–42_ for 22 h (**D**,**E**). Cell viability was assessed using the CCK-8 assay. Values are the mean ± SD from three independent experiments, and each one was conducted in triplicate. #: control group vs. Aβ group; *: Aβ + Se-GTEs group vs. Aβ group.; ### *p* < 0.001, * *p* < 0.05, ** *p* < 0.01, and *** *p* < 0.001, respectively.

**Table 1 foods-11-02159-t001:** Comparison of the main components of ESYL and ZYMJ (μg/g).

Chemical Constituents	ESYL (μg/g)	ZYMJ (μg/g)
Se	0.58 ± 0.01	4.33 ± 0.18
Tea polyphenols	16.74 ± 0.69	21.50 ± 0.47
Caffeine	4.43 ± 0.09	5.41 ± 0.31
Free amino acids	7.02 ± 0.03	6.76 ± 0.06
Soluble sugar	19.18 ± 1.07	11.65 ± 1.15
Water extracts	42.59 ± 0.16	46.23 ± 0.21

Values are means ± SD from three independent triplicate experiments.

## Data Availability

The data presented in this study are available upon request from the corresponding author. The data are not publicly available because of privacy.

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
