# Peer review of "Evaluation of the Brewing Characteristics, Digestion Profiles, and Neuroprotective Effects of Two Typical Se-Enriched Green Teas"

_foods, 2022, doi:10.3390/foods11142159_

Round 1

Reviewer 1 Report

General comments: The manuscript is very interesting and fits in the scope of the Foods journal. The manuscript provides information on the effects of the brewing method of traditional and Se-enriched tea on its antioxidant value and neuroprotective qualities. Some of the results the authors presented in the form of supplements, which is justified. I read the manuscript with real pleasure. However, I have a few comments.

Introduction: Is there a noted Se deficiency in the diet of Chinese people? Selenium is an essential element, but in excess it is toxic to the body and can, for example, contribute to the development of type II diabetes. It is important to mention.

Materials and methods: written appropriately

Results: should be changed to Results and Discussion, as there is currently no discussion mentioned in the manuscript, and it is essential for the paper to be considered scientific. This chapter contains a discussion of the results, but in sections 3.3. 3.4. and 3.5, a commentary should be written based on the available literature; besides, the discussion of the results is written appropriately, the authors explained the results obtained by citing the relevant literature

Table 1: ESST please change on ESYL

Figures: appropriate and readable, in Fig 1D and 1H, however, the title of the graph should be changed, now it is unclear, it should be noted that it is the number of tea brewing, not the brewing times

Author Response

Authors’ response

Dear editor and referees,

First of all, we would like to express our sincere thanks for your kind consideration and extremely valuable comments of titled manuscript “Evaluation for Brewing Characteristics, Digestion Profiles, and Neuroprotective Effects of Two Typical Se-Enriched Green Teas” (Manuscript ID: foods-1823327). We are pleased to follow the referees’ suggestions and revise the manuscript according to the comments. The specific modifications are shown below (changes to the manuscript are highlighted in red). Attached are the itemized responses to the referees’ reports, which also summarize the changes made to the manuscript. We hope you will find these responses and changes satisfactory.

If you have any other questions, please contact us: caijievip@whpu.edu.cn

Sincerely yours,

A/Prof. Jie CAI

July 17, 2022

Responses to the referees:

Reviewer #1: Comments and Suggestions for Authors

General comments: The manuscript is very interesting and fits in the scope of the Foods journal. The manuscript provides information on the effects of the brewing method of traditional and Se-enriched tea on its antioxidant value and neuroprotective qualities. Some of the results the authors presented in the form of supplements, which is justified. I read the manuscript with real pleasure. However, I have a few comments.

Comment 1: Introduction: Is there a noted Se deficiency in the diet of Chinese people? Selenium is an essential element, but in excess it is toxic to the body and can, for example, contribute to the development of type II diabetes. It is important to mention.

Response: Thanks a lot for your professional comment. Therefore, we have already made the following additions to the Introduction in the revised manuscript (Line 54-61): The recommended dietary allowance for Se intake in adults is currently 55 μg per day with a recommended tolerable upper intake level of 400 μg per day [1]. Inadequate dietary Se intake is detrimental to health and contributes to many diseases like cancer, coronary heart and Keshan disease [2], affecting 0.5-1 billion people worldwide [3]. However, excessive Se intake can induce toxicity on human body and lead to symptoms of alopecia, hair and nail abnormalities, skin injury, neurological disorders, even paralysis and death [4]. Thus, moderate Se supplementation is of great necessity for human health.

Comment 2: Materials and methods: written appropriately

Response: Thanks so much for your praise and affirmation of the section of Materials and methods for our manuscript.

Comment 3: Results: should be changed to Results and Discussion, as there is currently no discussion mentioned in the manuscript, and it is essential for the paper to be considered scientific. This chapter contains a discussion of the results, but in sections 3.3. 3.4. and 3.5, a commentary should be written based on the available literature; besides, the discussion of the results is written appropriately, the authors explained the results obtained by citing the relevant literature

Response: Thank you for your very constructive suggestion. We have already revised “Results” to “Results and Discussion” in the Line 241 as shown in the revised manuscript. For the section 3.3, 3.4 and 3.5, we have already added some more discussions to the revised manuscript (Line 307-309, Line 318-320, Line 333-334, Line 336-338, and Line 341-342).

Comment 4: Table 1: ESST please change on ESYL

Response: Thanks very much for your kind reminder. We have already revised “ESST” to “ESYL” in the Table 1 as shown in the revised manuscript.

Comment 5: Figures: appropriate and readable, in Fig 1D and 1H, however, the title of the graph should be changed, now it is unclear, it should be noted that it is the number of tea brewing, not the brewing times

Response: We appreciate this valuable advice. We have revised “brewing times” to “number of tea brewing” to make the graph clearer in Fig 1D and 1H as shown in the revised manuscript.

Reference

  1. Monsen, E.R. Dietary reference intakes for the antioxidant nutrients: vitamin C, vitamin E, selenium, and carotenoids. Journal of the American Dietetic Association 2000, 100, 637-640, doi:10.1016/s0002-8223(00)00189-9.
  2. Du, M.; Zhao, L.; Li, C.; Zhao, G.; Hu, X. Purification and characterization of a novel fungi Se-containing protein from Se-enriched Ganoderma Lucidum mushroom and its Se-dependent radical scavenging activity. European Food Research and Technology 2006, 224, 659-665, doi:10.1007/s00217-006-0355-4.
  3. Kieliszek, M.; Błażejak, S. Current knowledge on the importance of selenium in food for living organisms: a review. Molecules 2016, 21, 609, doi:10.3390/molecules21050609.
  4. Qin, H.B.; Zhu, J.M.; Liang, L.; Wang, M.S.; Su, H. The bioavailability of selenium and risk assessment for human selenium poisoning in high-Se areas, China. Environment International 2013, 52, 66-74, doi:10.1016/j.envint.2012.12.003.

Reviewer 2 Report

In the abstract the aim of the study should be better defined. Please reduce the numeber of keywords.

In the Introduction few information on phytochemical composition of green tea was reported. Please add numeric data for the most important bioactive compounds. Why is Se added to green tea in China?

In Materials&Methods information on raw materials should be added. Why did authors not use HPLC methods to identify the main polyphenolic compounds?

The Results should be better compare with similar previous studies. This section could be integrated with Discussion to avoid repetitions.

The conclusions should be better linked to the aim of the study.

Author Response

Authors’ response

Dear editor and referees,

First of all, we would like to express our sincere thanks for your kind consideration and extremely valuable comments of titled manuscript “Evaluation for Brewing Characteristics, Digestion Profiles, and Neuroprotective Effects of Two Typical Se-Enriched Green Teas” (Manuscript ID: foods-1823327). We are pleased to follow the referees’ suggestions and revise the manuscript according to the comments. The specific modifications are shown below (changes to the manuscript are highlighted in red). Attached are the itemized responses to the referees’ reports, which also summarize the changes made to the manuscript. We hope you will find these responses and changes satisfactory.

If you have any other questions, please contact us: caijievip@whpu.edu.cn

Sincerely yours,

A/Prof. Jie CAI

July 17, 2022

Responses to the referees:

Reviewer #1: Comments and Suggestions for Authors

Comment 1: In the abstract the aim of the study should be better defined. Please reduce the number of keywords.

Response: Thanks a lot for your professional comments. We have already re-written the aim of our study to make it better defined in the Abstract as shown in the revised manuscript (Line 22-25). In addition, we have already deleted the keywords “Brewing condition” in the Line 35 and modified the keywords “Neuroprotection” to “Neuroprotective effect” in the Line 36 as shown in the revised manuscript, and the number of keywords has been reduced from 6 to 5. We sincerely hope that the changes outlined above meet your expectations.

Comment 2: In the Introduction few information on phytochemical composition of green tea was reported. Please add numeric data for the most important bioactive compounds. Why is Se added to green tea in China?

Response: Thanks very much for your constructive comments. According to this proposal, we have added the following sentences (Line 48-52) in the first paragraph of the introduction section to properly introduce and discuss the most important bioactive compounds of tea: Especially, polyphenols, like catechins, are recognized as the most important bioactive compounds in various types of teas due to their key role in health-promoting and disease-preventing properties [1,2]. Furthermore, these polyphenols in green tea shows much higher level than that in black or oolong tea, which accounts for their antioxidant capacity [2].

For “Why is Se added to green tea in China?”, thank you for your question, and we sincerely hope the following explanation can make it clearer. Tea plant (Camellia sinensis) has strong enrichment ability for Se, and selenite is the main form of Se absorbed and utilized by tea plant [3]. In other words, the tea plant can absorb Se from the soil, and Se then accumulates in the leaves (Line 62-63). Previous studies have demonstrated that the presence of Se enhances the antioxidant, prebiotic, and anti-carcinogenic activity of tea [4].

Comment 3: In Materials & Methods information on raw materials should be added. Why did authors not use HPLC methods to identify the main polyphenolic compounds?

Response: Thanks so much for your advice, we have already supplemented the information of raw materials in the Materials & Methods section (Line 115-118) as shown in the revised manuscript. The detailed information of raw materials was added as follows: Enshi Yulu (ESYL) green tea was obtained from Enshi Lanbei Tea Co., Ltd. (Hubei Province, China), and Ziyang Maojian (ZYMJ) green tea was purchased from Ziyang Hongwei selenium-enriched Agricultural Technology Co., Ltd. (Shanxi Province, China), both with a date of manufacture in 2020.

Thanks for your question about “why did authors not use HPLC methods to identify the main polyphenolic compounds?”. In our work, China’s national standard methods (GB/T 8313-2018) was used for the determination of polyphenols in tea infusions. The traditional analytical methods including China’s national standard methods (GB/T 8313-2018) and spectrophotometry and high-performance liquid chromatography (HPLC) are widely used in the determination of tea polyphenols, and both of them show good accuracy [5]. Notably, spectrophotometry such as China’s national standard method (GB/T 8313-2018) is frequently used for the determination of the total polyphenols in tea samples, while HPLC is the main method for the simultaneous determination of individual tea polyphenols. The detection of total tea polyphenols can avoid the ambiguity, which is rather available without the need for sophisticated equipment. Despite HPLC had a satisfied separation performance and sensitivity, it requires relatively sophisticated and expensive analytical instruments. Given the consideration of the reasons described above and large sample size in our experiment, thus we selected China’s national standard method (GB/T 8313-2018) as the analytical methods of total tea polyphenols in tea infusions.

Comment 4: The Results should be better compared with similar previous studies. This section could be integrated with Discussion to avoid repetitions.

Response: We sincerely appreciate your suggestion. To be better compared with similar previous studies, we have already added some more discussions to the revised manuscript (Line 307-309, Line 318-320, Line 333-334, Line 336-338, and Line 341-342). We have also revised “Results” to “Results and Discussion” in the Line 241 as shown in the revised manuscript.

Comment 5: The conclusions should be better linked to the aim of the study.

Response: Thanks for your helpful comment. We have already carefully revised our conclusions to be better linked to the aim of the study. The corresponding changes can be found in the Conclusions section as shown in the revised manuscript.

Reference

  1. Wang, S.; Li, Z.; Ma, Y.; Liu, Y.; Lin, C.C.; Li, S.; Zhan, J.; Ho, C.T. Immunomodulatory effects of green tea polyphenols. Molecules 2021, 26, doi:10.3390/molecules26123755.
  2. Khan, N.; Mukhtar, H. Tea polyphenols in promotion of human health. Nutrients 2018, 11, doi:10.3390/nu11010039.
  3. Cao, D.; Liu, Y.; Ma, L.; Jin, X.; Guo, G.; Tan, R.; Liu, Z.; Zheng, L.; Ye, F.; Liu, W. Transcriptome analysis of differentially expressed genes involved in selenium accumulation in tea plant (Camellia sinensis). PLoS One 2018, 13, e0197506, doi:10.1371/journal.pone.0197506.
  4. Gao, Y.; Xu, Y.; Ruan, J.; Yin, J. Selenium affects the activity of black tea in preventing metabolic syndrome in high-fat diet-fed Sprague-Dawley rats. J Sci Food Agric 2020, 100, 225-234, doi:10.1002/jsfa.10027.
  5. Sun, M.F.; Jiang, C.L.; Kong, Y.S.; Luo, J.L.; Yin, P.; Guo, G.Y. Recent advances in analytical methods for determination of polyphenols in tea: a comprehensive review. Foods 2022, 11, doi:10.3390/foods11101425.

Reviewer 3 Report

The article is interesting, but it does contain some inaccuracies. They concern:

title: should be a bit shorter, in its current form it is quite illegible.

This is also related to the suggestion for keywords, which are too many and do not fully reflect the problem of the topic. In addition, they should be phrases, the most important keys that the reader can use to find the article in search engines. The authors are asked to correct these inaccuracies.

Abstract:

There is no information in this part of the article about what types of brewing were. Authors should present the characteristics of samples that were brewed differently. But it is not known how? There is no specific summary of the overall research that is the subject of this article.

Introduction: It is quite laconic and contains basic information, generally rather popular science. There are no significant issues that would make the reader meaningful to the experiment performed with respect to the relevant research on the scope of the topic.

The authors did not indicate the novelty of their experiment. How does the presented research differ from the previous ones that have already been published? Therefore, please outline the background around the analyzed raw material, especially since the authors later provide detailed information on the characteristics and occurrence and the possibility of their use.

However, there is no information that the research was aimed at a comparative analysis of different brewing processes at different times and at different temperatures, which in turn influenced the antioxidant activity of tea infusions. There is also no information about what substances were determined - this applies to amino acids.

Chapter: 2. Materials and Methods

Subsection: 2.2. Preparation of tea infusions

There is no information on the number of attempts and repetitions. This should be completed.

Subsection: 2.9. Statistical analysis

Authors must supplement with literature data, indicating the source.

Subsection: 3. Results

It is quite well run.

However, in Table 1, no statistics are available. The authors are asked to indicate the significance of the differences between the individual samples. Similarly, Figs. 1 - 5. The authors indicate no statistical differences. The question is, what was the purpose of the statistical analysis? It cannot be determined whether the examined factors had a significant influence or not. It needs to be completed.

Lack. There is a discussion of the results obtained with the results of other authors. It is very important in this type of manuscript. Please treat this suggestion especially, as discussing the results without trying to discuss it with the available ones. With the results, it complements and indicates the importance of the obtained results as well as their purposefulness.

Chapter: 4. Conclusions

The authors are asked to formulate 2-3 important conclusions that result from the experiment. Please also indicate the practical potential of the conducted research.

Chapter: References

Although the authors reached for many reports thematically related to the described results of the experiment, it should be clearly stated that these are quite old reports. In the available international literature there are many more recent reports related to the subject of the manuscript. Therefore, the authors need to refresh the literature a bit, so that it is not only historical, but also indicates the latest reports.

Author Response

Authors’ response

Dear editor and referees,

First of all, we would like to express our sincere thanks for your kind consideration and extremely valuable comments of titled manuscript “Evaluation for Brewing Characteristics, Digestion Profiles, and Neuroprotective Effects of Two Typical Se-Enriched Green Teas” (Manuscript ID: foods-1823327). We are pleased to follow the referees’ suggestions and revise the manuscript according to the comments. The specific modifications are shown below (changes to the manuscript are highlighted in red). Attached are the itemized responses to the referees’ reports, which also summarize the changes made to the manuscript. We hope you will find these responses and changes satisfactory.

If you have any other questions, please contact us: caijievip@whpu.edu.cn

Sincerely yours,

A/Prof. Jie CAI

July 17, 2022

Responses to the referees:

Reviewer #1: The article is interesting, but it does contain some inaccuracies. They concern:

Title: should be a bit shorter, in its current form it is quite illegible.

Response: Thanks a lot for your professional comments. We have already modified the title to “Evaluation for Brewing Characteristics, Digestion Profiles, and Neuroprotective Effects of Two Typical Se-Enriched Green Teas” instead of “Evaluation for Brewing Characteristics, Dynamic in Vitro Digestion Profiles, and Neuroprotective Effects of Two Typical Se-Enriched Green Teas” in the Line 2-4 as shown in the revised manuscript.

Keywords: This is also related to the suggestion for keywords, which are too many and do not fully reflect the problem of the topic. In addition, they should be phrases, the most important keys that the reader can use to find the article in search engines. The authors are asked to correct these inaccuracies.

Response: Thanks very much for your valuable comments. According to your suggestion, we have already deleted the keywords “Brewing condition” in the Line 35 and modified the keywords “Neuroprotection” to “Neuroprotective effect” in the Line 36 as shown in the revised manuscript, and the number of keywords has been reduced from 6 to 5. The revised Keywords are as follows: Se-enriched tea; Chemical component; Antioxidant activity; Dynamic in vitro digestion; Neuroprotective effect. We sincerely hope that the changes outlined above meet your expectations.

Abstract: There is no information in this part of the article about what types of brewing were. Authors should present the characteristics of samples that were brewed differently. But it is not known how? There is no specific summary of the overall research that is the subject of this article.

Response: Thanks very much for your kind reminder. We have already supplemented the types of brewing and the characteristics of samples that were brewed differently in the Line 25-29. And the specific summary of the overall research has been added in the Abstract section (Line 32-34). All of these changes above can be found in our revised manuscript.

Chapter: 1. Introduction

Comment 1: It is quite laconic and contains basic information, generally rather popular science. There are no significant issues that would make the reader meaningful to the experiment performed with respect to the relevant research on the scope of the topic.

Response: Thanks a lot for yours support on this work.

Comment 2: The authors did not indicate the novelty of their experiment. How does the presented research differ from the previous ones that have already been published? Therefore, please outline the background around the analyzed raw material, especially since the authors later provide detailed information on the characteristics and occurrence and the possibility of their use.

Response: Thank you so much for your helpful suggestion. According to these proposals, we have made extensive modification in the Line 48-52, Line 54-61, Line 77-80, Line 83-90, and Line 101-103 as shown in the revised manuscript. We sincerely hope that our modifications are satisfactory to you.

Comment 3: However, there is no information that the research was aimed at a comparative analysis of different brewing processes at different times and at different temperatures, which in turn influenced the antioxidant activity of tea infusions. There is also no information about what substances were determined - this applies to amino acids.

Response: Thanks a lot for your valuable comment, and we are sorry for the above lacking. We have supplemented the aim of comparative analysis of different brewing processes under different brewing conditions and the substances in the Line 104-107 as shown in the revised manuscript.

Chapter: 2. Materials and Methods

Comment 1: Subsection: 2.2. Preparation of tea infusions

There is no information on the number of attempts and repetitions. This should be completed.

Response: Thanks for your kind reminder. We have already supplemented the information on the number of attempts and repetitions in the revised manuscript (Line 160).

Comment 2: Subsection: 2.9. Statistical analysis

Authors must supplement with literature data, indicating the source.

Response: Thanks again for your kind reminder. We have supplemented the literature data in the Line 239-240 as shown in the revised manuscript. The changes made are as follows: Student’s t-test (P < 0.05) was used to compare the differences between groups [1].

Chapter: 3. Results

Comment 1: It is quite well run.

However, in Table 1, no statistics are available. The authors are asked to indicate the significance of the differences between the individual samples. Similarly, Figs. 1 - 5. The authors indicate no statistical differences. The question is, what was the purpose of the statistical analysis? It cannot be determined whether the examined factors had a significant influence or not. It needs to be completed.

Response: Thanks a lot for your suggestion. Significant difference is a statistical term. It is a statistical evaluation of data differences. Generally, when the experimental results reach the level of 0.05 or 0.01, there is a significant or extremely significant difference between the data. Typically, difference significance analyses are conducted using Student’s t-test (two group data) or one-way analysis of variance (ANOVA) (more than two group data) with P < 0.05 as significant. Data in Table 1 were presented as mean ± SD. Notably, Student’s t-test can not be conducted when there are only two group data but no control data. Thus, we believe that data in Table 1 is still available. In Figure 5, Student’s t-test was used to evaluated the significant effects (*P < 0.05, **P < 0.01 and ***P < 0.001, respectively).

Comment 2: Lack. There is a discussion of the results obtained with the results of other authors. It is very important in this type of manuscript. Please treat this suggestion especially, as discussing the results without trying to discuss it with the available ones. With the results, it complements and indicates the importance of the obtained results as well as their purposefulness.

Response: We appreciate your very constructive advice. We have already added some more discussions to the revised manuscript (Line 307-309, Line 318-320, Line 333-334, Line 336-338, and Line 341-342).

Chapter: 4. Conclusions

Comment 1: The authors are asked to formulate 2-3 important conclusions that result from the experiment. Please also indicate the practical potential of the conducted research.

Response: Thanks a lot for your suggestion. We have thought about this carefully and done our best to revise our important conclusions in light of these valuable comments. The corresponding changes can be found in the Conclusions section as shown in the revised manuscript (Line 435-438, Line 440-442, and Line 444-448). Moreover, the description of the practical potential of the conducted research has also been supplemented in the Line 452-454 as shown in the revised manuscript.

Chapter: References

Although the authors reached for many reports thematically related to the described results of the experiment, it should be clearly stated that these are quite old reports. In the available international literature, there are many more recent reports related to the subject of the manuscript. Therefore, the authors need to refresh the literature a bit, so that it is not only historical, but also indicates the latest reports.

Response: Thanks very much for your valuable comments. We have already updated the cited literatures with the latest works in the revised manuscript. We sincerely hope our modifications meet your expectations.

Reference

  1. Li, Z.; Zhu, A.; Song, Q.; Chen, H.Y.; Harmon, F.G.; Chen, Z.J. Temporal Regulation of the Metabolome and Proteome in Photosynthetic and Photorespiratory Pathways Contributes to Maize Heterosis. Plant Cell 2020, 32, 3706-3722, doi:10.1105/tpc.20.00320.
